# GNAQ Q209R Mutations Are Highly Specific for Circumscribed Choroidal Hemangioma

**DOI:** 10.3390/cancers11071031

**Published:** 2019-07-22

**Authors:** Claudia Helga Dorothee Le Guin, Klaus Alfred Metz, Stefan Horst Kreis, Nikolaos Emmanouel Bechrakis, Norbert Bornfeld, Michael Zeschnigk, Dietmar Rudolf Lohmann

**Affiliations:** 1Department of Ophthalmology, University Hospital Essen, University Duisburg-Essen, Hufelandstr. 55, 45147 Essen, Germany; 2Department of Pathology, University Hospital Essen, University Duisburg-Essen, Hufelandstr. 55, 45147 Essen, Germany; 3Institute of Human Genetics, University Hospital Essen, University Duisburg-Essen, Hufelandstr. 55, 45147 Essen, Germany

**Keywords:** circumscribed choroidal hemangioma, GNAQ, GNA11, oncogenic mutation, biopsy

## Abstract

Several tumors, including uveal melanoma, show somatic mutations of *GNAQ/GNA11*. Circumscribed choroidal hemangioma is a benign tumor that becomes symptomatic in adulthood. In some patients, morphologic examination of biopsies is required for differential diagnosis between amelanotic choroidal melanoma and circumscribed choroidal hemangioma. Here, we report the results of *GNAQ/GNA11* mutation analysis in samples from circumscribed choroidal hemangioma. Deep amplicon sequencing (Illumina MiSeq, San Diego, CA, USA) of positions R183 and Q209 of GNAQ and GNA11 in tissue samples from 33 patients with histologically diagnosed circumscribed choroidal hemangioma. All patients underwent biopsy or enucleation at our clinic between 2008 and 2018. To enable detection of variant alleles at low fractions, read depth exceeded 15,000-fold. DNA for genetic analysis was prepared from either snap-frozen (*n* = 22) or FFPE (*n* = 11) tissue samples. Samples from 28/33 patients (85%) showed a somatic missense mutation of *GNAQ* (c.626 A > G) predicted to result in p.Q209R. Variant allele fraction was variable (range 2.3% to 28%). Variants of *GNAQ* resulting in p.Q209 are characteristic for circumscribed choroidal hemangiomas. It appears that the *GNAQ* mutation spectrum in this tumor is narrow, possibly restricted to p.Q209R. Moreover, the spectrum is distinct from that of uveal melanoma, in which alterations resulting in p.Q209R are very rare.

## 1. Introduction

Circumscribed choroidal hemangioma (CCH) is a rare benign tumor that typically presents as a solitary orange-red mass, mostly temporal at the posterior pole of the fundus. It usually becomes symptomatic in adulthood (between the second and fourth decade) when subretinal or intraretinal fluid within the macula region or exudative retinal detachment cause visual problems [1,2,3]. In some cases, the tumor is detected during routine eye examination. Usually, diagnosis is established based on clinical findings. The results of fluorescein angiography (FLA) may be required to distinguish circumscribed choroidal hemangioma from other tumors like uveal melanoma or choroidal metastasis. FLA of CCH typically shows pre-arterial filling of the tumor vessels during the first few seconds of fluorescein angiography, while in late phase, a wash-out is observed [4].

Circumscribed choroidal hemangioma must be distinguished from diffuse choroidal hemangioma that is associated with Sturge-Weber-Syndrome (SWS), which is usually detected in childhood ipsilateral to the nevus flammeus. CCH has been observed in patients with SWS, but this appears to be rare [5].

Histomorphology of CCH is characterized by thin-walled vascular channels with endothelial cells and scant stroma cells [2]. Capillary malformations of similar morphology can also occur at other localizations such as skin. Oncogenic alterations of the *GNAQ* gene have been detected in endothelial cells within malformations at this location (p.R183Q, p.R183L, p.R183G in 8, 1, and 1 of 13 patients, respectively) [6]. It was postulated that *GNAQ* mutations in endothelial cells disturb the interaction between vascular and perivascular cells, which in turn contributes to the formation of capillary malformations with enlarged capillary lumens and ectatic venular morphology [6].

*GNAQ* and its paralogue *GNA11* each encode an alpha subunit of a heterotrimeric G protein, a binding protein that remains in its active GTP-bound state when mutated [7,8,9]. Both genes have been reported to be mutated in several vascular malformations such as congenital hemangioma of the skin and cherry angioma and in melanocytic tumors (choroidal nevus, uveal melanoma, melanocytoma, blue nevus) [6,9,10,11,12,13,14,15,16,17,18]. Moreover, *GNAQ* or *GNA11* mutations have been detected in several neural crest disorders such as Sturge Weber Syndrome (SWS). Interestingly, the various neoplastic entities show variable mutational profiles of *GNAQ* and *GNA11*. In uveal melanoma, mutations are frequent (>80%) and result in alterations at position Q209 or R183 of GNAQ or GNA11 [9]. A distinct profile is observed in affected tissues of SWS patients and of patients with non-syndromic port-wine stains where pathogenic mutations appear to be restricted to exon 4 of *GNAQ* [13,19,20,21,22,23].

The primary goals of the study presented here are (i) to determine if oncogenic alterations of *GNAQ/GNA11* are present in CCH and (ii) to obtain information on their mutational profile. It turned out that most CCH show oncogenic alterations and that the mutational profile of CCH is strictly distinct from that of other tumors at the same location, and results of genetic analysis can therefore provide evidence for the diagnosis of CCH.

## 2. Results

### 2.1. Clinical Characteristics

Between 2008 and 2018, tumor samples were obtained from 33 patients with final diagnosis of CCH. Initial diagnosis at the time of referral was uveal melanoma (*n* = 13, 39%), unknown tumor (*n* = 13, 39%), hemangioma (*n* = 5, 15%), metastasis (*n* = 1, 3%), or retinoma (*n* = 1, 3%) (Figure 1).

In most cases, histopathological examination of biopsy samples was performed to establish the diagnosis. In two hemangiomas with initially clear clinical diagnosis, biopsy was performed due to accelerated tumor growth. In a third patient with clinical diagnosis of hemangioma, the eye had to be enucleated for curative reasons.

The clinical features of all 33 CCH patients are presented in Table 1. The median age at diagnosis was 48.6 years, and median follow-up was 1.7 years (range 5 days–16.7 years) (Table 1). At last visit, visual acuity was better than 20/40 in *n* = 6 (18%), between 20/40 and 20/400 in n = 10 (30%), and 52% (*n* = 17) had a visual acuity below 20/400. In 24% (*n* = 8) a retinal detachment was present at initial presentation (also see Appendix A).

### 2.2. GNAQ/GNA11 Mutation Analysis

We performed Sanger and deep amplicon sequencing of *GNAQ* and *GNA11* specifically aimed at the positions coding for Q209 and R183 in DNA from all 33 tissue samples histologically diagnosed as CCH.

In 28 of the 33 CCH biopsy samples (85%), a c.626A>G, p.Q209R variant in exon 5 of the *GNAQ* gene was identified. In mutation positive samples, the variant allele fraction (VAF) as determined by quantitative deep amplicon sequencing was variable, ranging from 2% to 28% (median 19%, Appendix A). We could not detect any further mutation in the sequenced region in any sample. We confirmed the GNAQ p.Q209R mutations by Sanger sequencing using PCR primers located outside the first amplicon (GNAQ p.Q209long Appendix A) in all samples with a variant allele fraction >0.2. We also performed Sanger sequencing of candidate regions GNA14 p.R205 and GNAQ p.G48 in all but one sample without a GNAQ p.Q209R mutation but could not detect any mutation.

### 2.3. Bivariate Statistical Analysis

We compared clinical features of patients with GNAQ mutation (*n* = 28) and mutation negative tumors (*n* = 5). In the mutation positive patients, the median duration of symptoms, 42 days (range 3–1095 days), was shorter than in the mutation negative patients (287.5 days, range 4–730 days) (Figure 2a). Similarly, the median tumor height in the mutation positive tumors (3.13 mm, range 1.39–5.2 mm) was smaller than in the negative tumors (4.3 mm, range 3–5.89 mm) (Figure 2b).

In contrast, neither the median largest basal tumor diameter LBD) nor the median smallest basal tumor diameter (SBD) showed a clear difference between the two groups (Figure 2c and not shown).

Linear regression of tumor volume on variant allele fraction (VAF) showed a positive correlation (*p* = 0.0179, see Figure 3), whereas age at diagnosis was negatively correlated with the VAF (*p* = 0.000659) (Figure 4). Duration of symptoms, which is the time period reported by the patient at initial presentation during which e.g., visual disturbances have been observed, is not associated with age at diagnosis (Appendix A).

## 3. Discussion

### 3.1. Selection Bias

Circumscribed choroidal hemangioma (CCH) is a rare neoplasm. Most often clinical criteria are sufficient to establish diagnosis of CCH. In a few cases, histopathological examination of tissue samples is necessary to make correct diagnosis. For this reason, most publications on CCH are limited to clinical data. Here we present molecular genetic data on 33 CCH cases in which tumor tissue samples were taken for diagnostic or therapeutic reasons (32 and 1 patients, respectively). Although these selection criteria might introduce some bias, this might not necessarily have a negative effect on the validity of the findings with respect to patients with CCH in general. The clinical features of patients in our set of CCH cases were similar to those of published cohorts. For example, the median age at diagnosis in our 33 patients at 48 years was within the range of published CCH cohorts with 39, 45, and 57 years [2,3,22]. In agreement with these three reports, localization of most CCHs in our cohort (approximately 73%) was temporal. Tumors in our study showed a larger median height and larger median largest basal diameter compared to data by Krohn et al. [24] (median tumor height 2.2 mm, median largest basal diameter 7.5 mm) and Shields et al. [2] (median height 3 mm; median largest basal diameter 6 mm). In agreement with the findings reported by Witschel et al., all CCH in our study were less than 6 mm in height [3]. These comparisons of clinical features do not raise the suspicion that our set of CCH cases fails to be representative for patients with CCH in general.

### 3.2. No Variant Identified in 5 Samples

Samples from hemangioma and other neoplastic vascular malformations show variable proportions of neoplastic cells. Using a method with a low limit of detection (<1% VAF), we identified oncogenic alterations in 28 of 33 samples (%) with variant allele fraction (VAFs) ranging from 2% to 28%. In the remaining five samples, the signals for oncogenic alleles at *GNAQ/GNA11* position Q209 or R183 were within the range of noise. Therefore, it is unlikely that these samples have an oncogenic alteration at the sites covered by our assay. However, this does not exclude that, in some of these five samples, oncogenic activation of a G protein alpha subunit is present because of an alteration at another position of *GNAQ/GNA11* or in a gene coding for another subunit, e.g., *GNA14* [25].

### 3.3. Restricted Mutation Spectrum

All 28 samples showed the identical variant, *GNAQ c.626A > G* (p.Q209R). This is remarkable in two ways. On the one hand, mutational activation in CCH appears to be restricted to one specific amino acid substitution (Q > R) at one site (p.Q209) in only one of the G protein alpha subunit genes (*GNAQ*), whereas other neoplasms with mutant *GNAQ/GNA11* typically show some variation with respect to the kind and site of substitution and the gene. This result is supported by a recent report of Francis et al., who performed *GNAQ/GNA11* mutation analysis in a series of neoplasms including two CCH samples [13]. In Table 2 of this paper, it is stated that both samples show a GNAQ p.Q209R variant. The second remarkable aspect is that this variant is rarely observed in other neoplasms with mutant *GNAQ/GNA11*. For example, data available in COSMIC (https://cancer.sanger.ac.uk/cosmic) show the presence of GNAQ p.Q209R in only 8/850 melanomas of the eye.

### 3.4. Mutation and Phenotype

Hemangiomas are distinct from solid tumors such as uveal melanoma because they are composed of diverse cell types, many of which are genetically normal. The proportion of mutant cells in samples from our cohort appeared to be highly variable, ranging from 5% to 56%, as deduced from the VAF. The observation that the proportion of tumor cells, defined as those cells carrying the mutation, increases with tumor volume (Figure 3) may be due to an increased proportion of normal cell that contaminate the sample in particular in biopsy of small tumors. Whether other properties of hemangioma itself, such as vessel size or vessel wall thickness, contribute to the different VAFs has yet to be investigated.

Interestingly, the median age at diagnosis of CCH patients with *GNAQ* mutation, 50 years (range 27–77 years) was higher than that of the patients of the mutation negative cohort (35 years, range 7–58 years, Figure 2d). In addition, duration of symptoms was shorter, and the tumor size was smaller in the mutation positive cases (Figure 2). This suggests that mutation positive and negative hemangiomas have a different phenotype and that their pathogenesis might thus be distinct. However, as the cohort of mutation negative patients in our study are rather small this needs to be validated.

#### 3.4.1. Mutation Profile

Differences in mutational profiles between tumor entities are common and are likely to result from differential selection of distinct oncogenic variants. In an attempt to relate similarities between *GNAQ*/*GNA11* mutation profiles to different tumor entities, we performed unsupervised hierarchical cluster analysis (Figure 5). When interpreting the results, which are summarized in a dendrogram, it has to be kept in mind that, as clustering is not an entirely objective process, any conclusions are in part subjective. From the dendrogram, it is possible to discern four clusters.

Tumor entities that contribute to the cluster that is dominated by variants at GNAQ p.R183 are congenital tumors/vascular malformations such as port wine stains, portwine macrocheilia and Sturge-Weber Syndrome, non-Sturge Weber capillary malformations, and diffuse choroidal hemangioma, as well as phakomatosis pigmentvascularis [13,19,20,21,22,23,26,27,28]. These malformations/tumors are diagnosed early in life, and patients often show somatic mosaicism for the variant alleles in non-neoplastic cells. A disorganization of perivascular cells is seen in these congenital tumors/vascular malformations instead of cell proliferation. Somatic mosaicism implies that the underlying mutation (mostly R183) has occurred prenatally during embryonic development in undifferentiated cells. From the phenotype observed in patients, it appears that these cells have a limited proliferative activity in adulthood. By contrasts, tumor entities that contribute to the second and third main clusters, which is predominated by alterations resulting in GNAQ or GNA11 p.Q209 (indicated by grey bars), mostly show later age at diagnosis and therefore are presumably initiated during adulthood. Congenital hemangioma of the skin (RICH and NICH), which also cluster within this group, are an exception [10]. Other neoplasms that belong to this cluster are melanocytic tumors, specifically melanocytoma, nevus of the skin, blue nevus, choroidal nevus, choroidal melanoma, and vascular tumors like congenital hemangioma, hepatic small vessel neoplasia, anastomosing hemangioma, cherry angioma, and circumscribed choroidal hemangioma [9,10,12,13,14,15,16,17,18,29,30,31,32,33]. It is common for these tumors that make up this cluster to show continuous cell proliferation.

The fourth cluster (black bar) is characterized by mutations resulting in GNAQ p.Q209 only (exception cherry angioma with one R183G mutation). The GNA-variant profile of CCHs is most similar to that of cherry angioma, anastomosing hemangioma, and hepatic small vessel proliferation. In these acquired tumors, only GNAQ p.Q209H and p.Q209R mutations have been observed. In this group of tumors, CCHs occupy a prominent position because they exclusively exhibit Q209R mutations. Interestingly, hepatic small vessel proliferation, which are part of the GNAQ p.Q209 cluster, can also show variants in *GNA14*. Therefore, this gene is an interesting candidate for mutation analysis in variant negative CCHs.

#### 3.4.2. Differential Selection of Distinct Oncogenic Variants

Using in vitro assays, it was shown that the different oncogenic GNA variants have different functional consequences. For example, variants affecting p.R183 activate the p38 MAP Kinase pathway, whereas those at p.Q209 in addition activate c-Jun N-terminal kinase (JNK) and ERK [28]. In other words, the spectrum of oncogenic actions that are a consequence of p.R183 alterations appears to be narrower than that of p.Q209 changes. This difference in oncogenic actions may in part explain why somatic mosaicism has been observed only for p.R183 mutations. In other words, p.R183 alterations, but not p.Q209 alterations, are tolerated in non-neoplastic cells of patients. One interpretation of this finding is selection bias against the presence of p.Q209 alterations because cells with p.Q209 alterations cannot form a mutant sector of non-neoplastic cells. However, for neoplastic cells the broader spectrum of downstream effects mediated by p.Q209 alterations is of selective advantage and this is also reflected by the predominance of p.Q209 alterations in neoplasms of the adult.

Based on the data presented here, it is questionable whether CCHs result from congenital changes as previously suggested or from somatic mutations in adult organisms [34,35,36].

## 4. Materials and Methods

### 4.1. Patients

All except one sample were obtained by biopsy that was performed for diagnostic purpose. In a single case, the affected eye was treated by enucleation. Transretinal tumor biopsy was performed as previously described by Akgul et al. [37].

### 4.2. Sanger Sequencing

Exon 5 sequences of *GNAQ* and *GNA11* were determined by PCR-based capillary Sanger sequencing using tagged primers for PCR. Sequences of oligonucleotides (Eurogentec, Liège, Belgium) are given in Table 1, with the 5′universal Tag sequence highlighted in capital letters. Different universal tags were used for forward and reverse primers. PCR was performed in a reaction volume of 25 µL containing 10–20 ng (1–2 ng of FFPE) genomic DNA, 12.5 µL of Q5 High Fidelity 2 × Master mix (New England Biolabs, Frankfurt, Germany), and 0,1 µM of each primer. The thermal cycling profile was the following: Initial denaturation at 98 °C for 30 s, followed by 35 rounds of amplification at 98 °C for 10 s, 58 °C for 30 s, and 72 °C for 30 s. A final extension step at 72 °C for 2 min was added. The PCR products were purified using ExoSAP-IT (Affymetrix, Santa Clara, CA, USA) according to the manufacturer’s protocol. Sequencing of the PCR products was carried out using the BigDye Terminator v1.1 Cycle Sequencing Kit (Applied Biosystems, Foster City, CA, USA) following the manufacturer’s instructions with the sequencing primer (0,1 µM) hybridizing to the Tag sequence of the PCR primer. Automated electrophoreses was performed on Genetic Analyzer ABI PRISM 3130xl (Applied Biosystems). Sequencing data were analyzed using Geneious Pro (5.6.4.) software (Biomatters Ltd., Auckland, New Zealand). For validation of GNAQ p.Q209R mutation, we repeated the PCR using a second primer set located outside the first amplicon (GNAQ Q209long, Appendix A).

### 4.3. Deep Amplicon Sequencing

For quantitative analysis of the variant allele fractions (VAF), we performed deep amplicon sequencing on a next-generation sequencing (NGS) platform (Illumina MiSeq) using the same target specific primers and conditions for first round PCR as described above. In a second round PCR, 1 µL first round un-purified PCR product was amplified to attach Illumina specific adapter sequences and indexes (i5 NSE501Ftag and i7 N701Rtag, 0,1µM each, Appendix A) under the following cycling conditions: 98 °C 30 s, 15× (98 °C, 10 s; 53° C, 20 s; 72 °C, 30 s), 72 °C 10 min. Purification of PCR product is performed with Agencourt AMPure beads XP according to the manufacturer’s instructions. NGS sequencing was performed on a MiSeq paired-end run (2 × 150bp) following the manufacturer’s instructions. After MiSeq cluster generation, sequencing, and primary analysis, which includes base calling, filtering, and quality scoring, demultiplexed GNU-compressed FASTQ files (separated by index) are generated. For each index, a R1 and R2 *.fastq.gz file is obtained containing separate forward and reverse reads of the same sample.

### 4.4. Analysis of Sequencing Read Data

We used a python (Version 2.7.14) script to parse sequencing read data and determine variant allele fractions (VAF = number of variant reads at a nucleotide position/total number of reads covering this position). In brief, reads specific for *GNA11* and *GNAQ* were separated and trimmed to a string of 20 bases including the region of interest (R183 and Q209). To remove potential erroneous calls due to the sequencing process, mismatched R1-R2 read-pairs were discarded. However, even for matching R1– and R2–read pairs, each of the two times 20 bases had to meet a quality score of 35 or higher. In the final step, the number of occurrences of the various variants and the number of normal reads was counted.

To prove the quantitative nature of deep amplicon sequencing technology, we have generated a standard curve by analyzing various ratios of mutant and normal alleles adjusted by mixing tumor DNA with a heterozygous *GNAQ* c.626A>T mutation and normal DNA (see Appendix A). For all measurements, 10 ng of DNA was used for the first round PCR. In reads amplified from normal DNA samples, variant alleles at the region of interest occurred in a frequency of up to 0.02%, which is to be considered as detection limit. Comparing the VAF of normal DNA (0.02%) with the VAF range of the CCH samples (2.2–28%), we could see a bimodal distribution pattern with a gap of more than factor 50 between mutation positive and negative (norm) cases. Therefore, noise values are well separated from mutant reads (also see Appendix A).

### 4.5. Statistical Analysis

Statistical analysis and graphical visualization of the results was performed using the statistical programming software environment R (R Version 3.6.0, https://www.r-project.org/) and the ggplot2 package. In R, hierarchical agglomerative cluster analysis of mutational profiles was performed with the hclust function (stats package) using the ward D method. Results were plotted using the R package dendextend (https://doi.org/10.1093/bioinformatics/btv428).

## 5. Conclusions

The narrow mutation spectrum of CCH, which is restricted to GNAQ p.Q209R, supports us in establishing the diagnosis of CCH. This is of particular benefit in cases where not enough tissue sample is available for histological diagnosis.

## Figures and Tables

**Figure 1 cancers-11-01031-f001:**
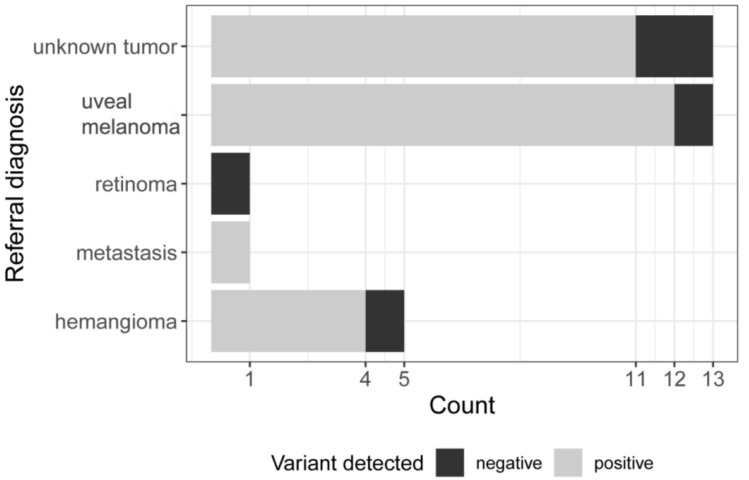
Referral diagnosis. Horizontal bars indicate the number of different referral diagnoses. Black: negative—no variant detected. Grey: positive—GNAQ p.Q209 variant detected.

**Figure 2 cancers-11-01031-f002:**
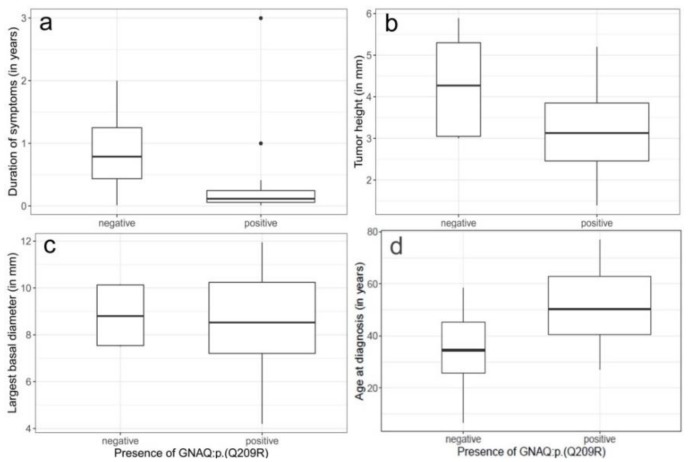
(**a**–**d**) Distribution of clinical characteristics between the mutation negative and mutation positive cohort. (**a**) duration of symptoms, (**b**) tumor height, (**c**) largest basal diameter, and (**d**) age at diagnosis. Boxplots with median represented as a solid line. Rectangles show the interquartile range. Width of the rectangles corresponds to the number of patients. None of the tested correlations is statistically significant.

**Figure 3 cancers-11-01031-f003:**
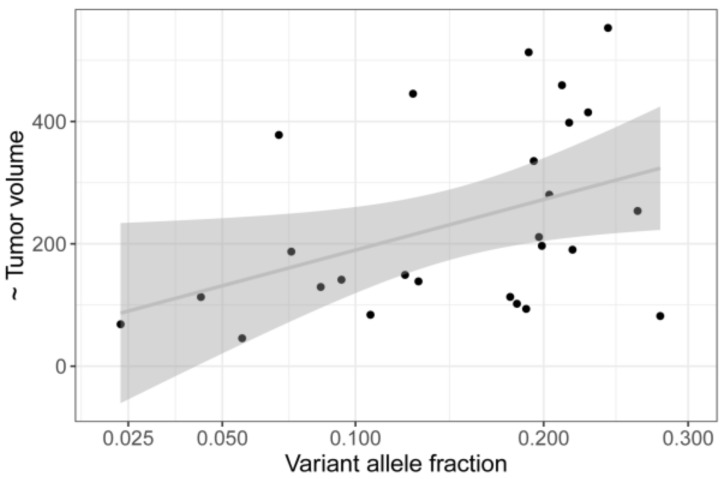
Linear regression of tumor volume on variant allele fraction (VAF) in 28 mutation positive samples. There is a trend of larger tumors showing higher VAF (*p* = 0,0179).

**Figure 4 cancers-11-01031-f004:**
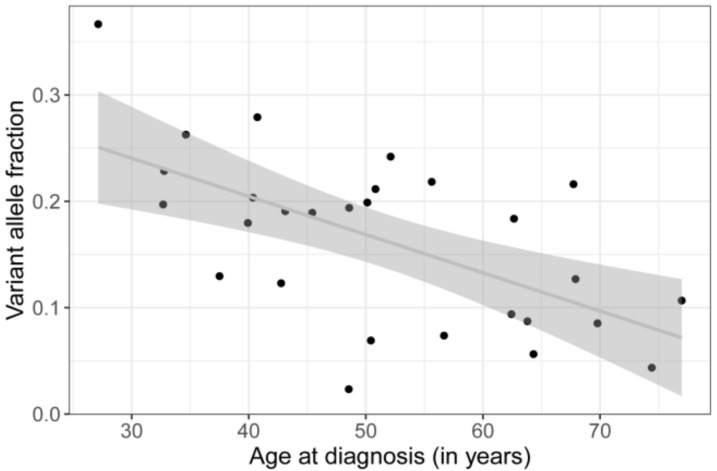
Linear regression analysis shows a significant negative correlation of patients age and VAF (*p* = 0.000659).

**Figure 5 cancers-11-01031-f005:**
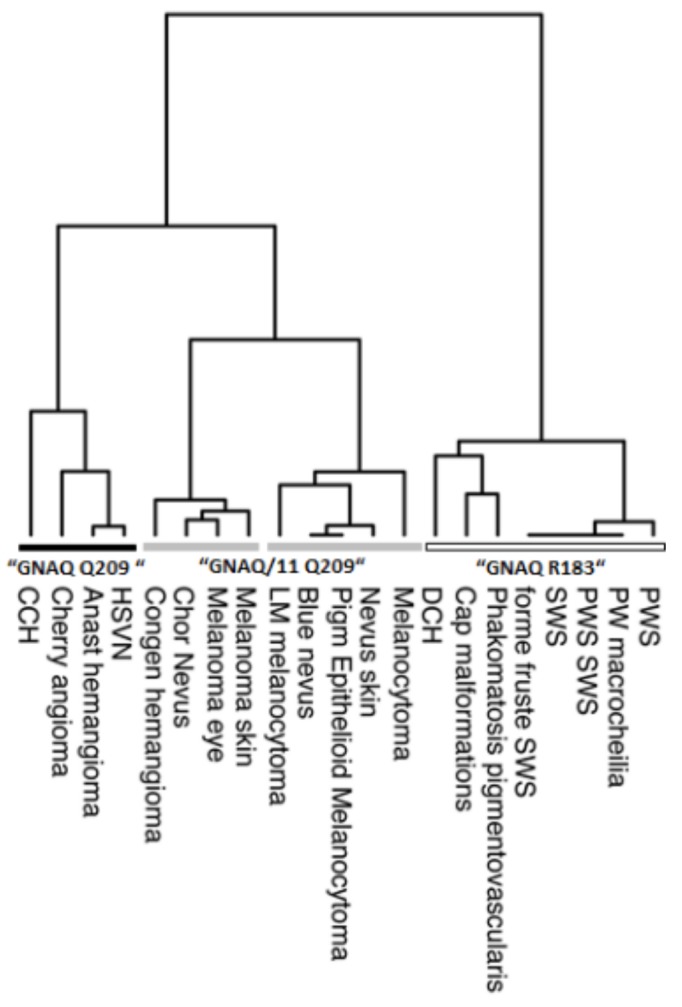
Unsupervised hierarchial cluster analysis of the *GNAQ/GNA11* mutation profile in different tumor entities. Four main clusters are formed which are named after the predominant mutation. Analysis is based on data presented in this paper and elsewhere (see Appendix A).

**Table 1 cancers-11-01031-t001:** Clinical characteristics of 33 CCH patients.

Clinical Features	All Patients (*n* = 33) or Proportion of Patients
Median Age at diagnosis (years)	49 (6.9–72)
Median observation period (years)	1.7 (0.013–16.7)
Median duration of symptoms (days)	156 (3–1095)
Tumor height (mm)	3.3 (1.39–5.89)
LBD (mm)	8.7 (4.2–11.9)
SBD (mm)	7.2 (3.1–10.4)
Laterality OD	20 (61%)
Female sex	12 (36%)
location	
nasal	7 (21%)
superior	2 (6%)
temporal	24 (73%)
juxtapapillary	17 (52%)
juxtamacular/submacular	14 (42%)
Retinal detachment	8 (24%)
Fundus pigmented (yes)	15 (45%)
Tumor pigmented (yes)	7 (21%)
Initial symptoms	
Decrease of visual acuity	22 (67%)
Metamorphopsia	1 (3%)
blurred vision	8 (24%)
flashes	6 (18%)
visual field defects	6 (18%)
Visual acuity initial	
>20/40	8 (24%)
<20/40> 20/400	17(52%)
<20/400	8 (24%)
Visual acuity last	
>20/40	6 (18%)
<20/40> 20/400	10 (30%)
<20/400	17 (52%)

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
