# Peer review of "GNAQ Q209R Mutations Are Highly Specific for Circumscribed Choroidal Hemangioma"

_cancers, 2019, doi:10.3390/cancers11071031_

Round 1

Reviewer 1 Report

Well designed and written small study which provides benefit for diagnosis of CHH.

Some comments and suggestions for corrections:

- please provide ethical approval

- the order of supplemental figures should be checked

- please explain in a short sentence how the duration of symptoms is defined

- please explain the abbreviations LBD and SBD in the text

Author Response

Dear Editor, dear reviewer,

included is the revision of the manuscript „GNAQ Q209R mutations are highly specific for circumscribed choroidal hemangioma”

Changes that have been made are listed point by point and are highlighted in yellow in the revised version of the manuscript:

Reviewer 1

Well designed and written small study which provides benefit for diagnosis of CHH.

Some comments and suggestions for corrections:

- please provide ethical approval

Authors’ comment:

Although, formally, an ethical approval is not mandatory for such a type of study, we have applied for such an approval. We are currently awaiting approval but were informed that this may take time.

- the order of supplemental figures should be checked

Authors’ comment:

We thank the reviewer. The order of the supplementary figures is correct but the labeling was wrong, therefore supplementary Figure 3 was renamed to supplementary Figure 2 and vice versa.

- please explain in a short sentence how the duration of symptoms is defined

Authors’ comment:

 â€śthe time period reported by the patient at initial presentation during which e.g.visual disturbances have been observed” is now added at line 113 when first mentioned in the text.).

- please explain the abbreviations LBD and SBD in the text (line 108)

Authors’ comment:

Abbreviations have now been explained in the text (line 108).

Reviewer 2 Report

By using deep amplicon sequencing of position R183 and Q209 of GNAQ and GNA11 in 33 circumscribed choroidal hemangiomas (CCH) , Guin et al  identified 28 out of 33 CCHs harbor GNAQ Q209R mutations. And then they concluded that GNAQ Q209R mutations are highly specific for CCH. Their findings are very interesting and novel. However, I have below comments.

1.    In tables 1, the authors show that 7 tumors out of 33 CCHs are pigmented. It is not clear why these CCH tumors are pigmented. Usually, some uveal melanomas are pigmented. Can authors give explanation? 

2.    In this paper, the authors mentioned that no GNAQ/11 variants were identified in 5 samples. Because only R183 and Q209 positions of GNAQ/11  were examined, the authors should also check other positions such as G48. GNAQ G48L mutation has been found in Hepatic small vessel neoplasms. 

3.    In fig.5, the authors should include details for each disease such as  numbers of samples as well as what kind of mutations.  

4.    Some reference citations are not precise. For example, In section “differential selection of distinct oncogenic variants”, when the authors mentioned “variants affecting p.R183 activate the p38 MAP kinase pathway whereas those at p.Q209 in addition activate c-Jun N-terminal kinase (JNK) and ERK “, they cited references 14 and 34. However, these two references do not have direct data that supports this claim. The authors should cite original references.

Author Response

Dear Editor, dear reviewer,

 included is the revision of the manuscript „GNAQ Q209R mutations are highly specific for circumscribed choroidal hemangioma”

Changes that have been made are listed point by point and are highlighted in yellow in the revised version of the manuscript:

Reviewer 2

By using deep amplicon sequencing of position R183 and Q209 of GNAQ and GNA11 in 33 circumscribed choroidal hemangiomas (CCH), Guin et al.,  identified 28 out of 33 CCHs harbor GNAQ Q209R mutations. And then they concluded that GNAQ Q209R mutations are highly specific for CCH. Their findings are very interesting and novel. However, I have below comments. 

1.   In tables 1, the authors show that 7 tumors out of 33 CCHs are pigmented. It is not clear why these CCH tumors are pigmented. Usually, some uveal melanomas are pigmented. Can authors give explanation? 

Authors’ comment:

Fundus pigmentation varies between individuals. In case of a darker pigmented fundus, the hemangioma appears lighter and vice versa. In some cases hemangiomas appear pigmented especially at the margins most likely due to compression of the choroid.

2.   In this paper, the authors mentioned that no GNAQ/11 variants were identified in 5 samples. Because only R183 and Q209 positions of GNAQ/11 were examined, the authors should also check other positions such as G48. GNAQ G48L mutation has been found in Hepatic small vessel neoplasms. 

Authors’ comment:

We have performed targeted GNAQ G48 and GNA14 R205 sequencing on all 4 out of 5 samples without GNAQ Q209 mutation. DNA of the fifth sample was no more available. The Sequences of the four samples did not give any hint for mutations in the analyzed regions.

3.   In fig.5, the authors should include details for each disease such as numbers of samples as well as what kind of mutations.

Authors’ comment:

We present the requested data now in an additional table (supplementary table 2).

4.   Some reference citations are not precise. For example, In section “differential selection of distinct oncogenic variants”, when the authors mentioned “variants affecting p.R183 activate the p38 MAP kinase pathway whereas those at p.Q209 in addition activate c-Jun N-terminal kinase (JNK) and ERK “, they cited references 14 and 34. However, these two references do not have direct data that supports this claim. The authors should cite original references

Authors’ comment:

We thank the reviewers for carefully evaluating the manuscript. The incorrectly cited references 14 and 34 are now replaced by the correct reference which is Thomas et al., (ref. 33).

Reviewer 3 Report

The authors describe performed targeted sequencing on CCH, a very uncommon type of benign tumor and describe GNAQ Q209R mutations. This study has some intrinsic strengths and weaknesses as well as possible improvements.

Strengths: 

large sample size of this rare tumor type. 

Though targeted sequencing, most the mutational driver was found in most tumors. 

GNAQ Q209R is rare in cancers, no mutations in GNA11 are all thought provoking and authorshad a nice discussion

Weaknesses: 

GNAQ/11 mutations are now well described in vascular tumors and the Q209R mutation have also been described, limited the novelty/impact somewhat

Targeted sequencing of two genes today, with the increased affordibility of whole exome sequencing, also limits impact.

Suggestions

One important potential technical pitfall that can cause false positives is PCR product contamination, especially if the same primers are used for all sequencing. Authors should validate finding with another pair of primers with one that is outside of the current amplicon--just Sanger and 1-2 samples with high VAF is sufficient.

The exact base change (c.626A->G) should be abstract and results

Difference is mutational preference can be functional or be due to different mutational processes (UV vs carcinogen for example). Mutational processes can explain the difference between specific codon changes between tumor types of the same gene such as p53. Is teh c.626A->G part of a mutational process.

Figure 2 doesn't have p-value stats

Looked up refs 14 and 34 and cannot find mention of Q209 and R183 activating different pathways.

Author Response

Dear Editor, dear reviewer,

included is the revision of the manuscript „GNAQ Q209R mutations are highly specific for circumscribed choroidal hemangioma”

Changes that have been made are listed point by point and are highlighted in yellow in the revised version of the manuscript:

Reviewer 3

The authors describe performed targeted sequencing on CCH, a very uncommon type of benign tumor and describe GNAQ Q209R mutations. This study has some intrinsic strengths and weaknesses as well as possible improvements.

Strengths: 

large sample size of this rare tumor type. 

Though targeted sequencing, most the mutational driver was found in most tumors. 

GNAQ Q209R is rare in cancers, no mutations in GNA11 are all thought provoking and authors had a nice discussion

Weaknesses: 

GNAQ/11 mutations are now well described in vascular tumors and the Q209R mutation have also been described, limited the novelty/impact somewhat

Authors’ comment:

It is correct that GNAQ/11 mutations are described in vascular tumors, but mostly a variety of different mutations are detected for the same tumor type. (For example GNA11 Q209L, GNAQ Q209H,GNAQ Q209L,GNAQ Q209P mutations in congential hemangioma (various locations)(Ayturk et al 2016).

Here we describe for the first time that GNAQ Q209R mutations are highly specific for circumscribed choroidal hemangioma. Such a strong statement could not be made on the basis of the data published previously, as the number of samples examined was insufficient. E.g. in publication by Francis et al. 2018 the number of solitary choroidal hemangiomas was just n=6. Even more, the presentation of the results in this publication is inconsistent with the abstract as well as the results section reporting a GNAQ c.626A>T (p.Gln209Leu) mutation in 100% of the solitary choroidal hemangiomas but contradictory reporting GNAQ p.Q209R mutations in Table 1 (Francis et al.).

Since c.626A>T;p.Gln209Leu is frequently observed in uveal melanoma, the specificity of GNAQ Q209R mutations for solitary choroidal hemangioma could not be observed in this study.

Targeted sequencing of two genes today, with the increased affordability of whole exome sequencing, also limits impact.

Authors’ comment:

The low amount and quality (FFPE in some samples) of tissue (biopsy) available for DNA isolation and genetic analysis does not allow for deep exome sequencing in most samples. Furthermore, our main findings do not depend on witch of the numerous suitable technologies is used.

Suggestions

One important potential technical pitfall that can cause false positives is PCR product contamination, especially if the same primers are used for all sequencing. Authors should validate finding with another pair of primers with one that is outside of the current amplicon--just Sanger and 1-2 samples with high VAF is sufficient.

Authors’ comment:

We followed the suggestion of the reviewer and performed targeted Sanger sequencing using primers located outside the current amplicon on 6 samples with an variant allele fraction >0.2. Mutations were confirmed in all samples.

The exact base change (c.626A->G) should be abstract and results

Authors’ comment:

The sequence variation is now described in the abstract line 25 and results line 93.

Difference is mutational preference can be functional or be due to different mutational processes (UV vs carcinogen for example). Mutational processes can explain the difference between specific codon changes between tumor types of the same gene such as p53. Is the c.626A->G part of a mutational process.

Authors’ comment:

We had looked into this. Specifically, we checked if different mutational processes might explain (part) of the differences in mutation profiles. It appeared that the specific alteration (c.626A->G) is compatible with several mutational processes and, therefore, this specific alteration is not informative in this respect.

Figure 2 doesn't have p-value stats

Authors’ comment:

None of the tested correlations is statistically significant. This is now mentioned in the figure legend.

Looked up refs 14 and 34 and cannot find mention of Q209 and R183 activating different pathways.

Authors’ comment:

We thank the reviewers for carefully evaluating the manuscript. The incorrectly cited references (Ref 14 and 34) are now replaced by the correct reference which is Thomas et al., (ref. 33).

Round 2

Reviewer 3 Report

The authors have adequately addressed my concerns.